# Modeling and analysis of UAV charging scheduling in fixed/mobile charging station systems

1st Zeyu Guo
*School of Mathematics*
*Southeast University*
Nanjing, China
eyuguoll@163.com

2nd Sining Zhang
*China north vehicle research institute*
Beijing, China
13426157603@163.com

3rd Jiahe Wang
*School of Mathematics*
*Southeast University*
Nanjing, China
213220649@seu.edu.cn

4th Xinyuan Huang
*School of Mathematics*
*Southeast University*
Nanjing, China
213220199@seu.edu.cn

5th Ruixu Hu
*School of Mathematics*
*Southeast University*
Nanjing, China
220231953@seu.edu.cn

6th Wenying Xu
*School of Mathematics*
*Southeast University*
Nanjing, China
wyxu@seu.edu.cn

*Abstract*—This paper proposes two novel mathematical models for optimizing the charging schedules of Unmanned Aerial Vehicles (UAVs) within systems featuring either fixed or mobile charging stations. The primary objective is to minimize the total charging time for all UAVs. Initially, the fixed charging station (FCS) system is modeled, followed by a comparison of four different algorithms. Subsequently, the model is extended to consider a mobile charging station (MCS) system, where the station can relocate as necessary. In this scenario, an algorithm is proposed to optimize the charging station's position to enhance charging efficiency. Finally, a numerical example is presented to compare the performance of different algorithms in both fixed and mobile charging station systems. The simulation results demonstrate that the proposed algorithm effectively improves charging efficiency by strategically positioning the charging station.

*Index Terms*—Unmanned aerial vehicles, charging scheduling, fixed charging station, mobile charging station

## I. Introduction

In recent years, unmanned aerial vehicles have been widely used in military, civil and commercial fields, showing great potential for application. In the military field, UAVs are widely used for tasks such as dangerous reconnaissance [1], search and rescue [2], and logistics distribution [3] due to their flexible deployment capability. Compared with ordinary vehicles, UAVs can quickly reach areas with complex road conditions, greatly improving the efficiency of rescue and cargo transportation missions [4]. Due to the limitations of battery technology and longer mission times, the endurance issue of UAVs has become a major barrier to their development. To overcome this problem, there has been some research to improve the battery technology of UAVs. Reference [5] summarizes recent advancements in battery technology. Another way to enhance endurance is to plan the scheduling of UAVs such as path planning and task assignment, which is a strategy used in many studies [6]–[9]. However, these studies have some shortcomings. On one hand, some research imposes

restrictions on the charging scenarios, such as requiring mobile charging vehicles to travel to the drones for power supply [8]. In certain situations, UAVs may need to perform hazardous tasks in remote areas where charging vehicles cannot reach. This requires UAVs to travel to the charging vehicle location for energy replenishment. On the other hand, these studies do not pay enough attention to the queuing problem caused by the increased demand for charging, as their focus is on the path and task planning of drones. Therefore, it is essential to study the queuing problem in the scenario where UAVs need to travel to charging stations to recharge their batteries.

The charging scheduling problem for commercial electric vehicles shares similarities with that of drones [10]. The former has conducted thorough research on the queuing problem in the charging process, providing methods and insights for the latter. Based on the mobility of charging stations, these studies can be categorized into fixed charging stations (FCS) and mobile charging stations (MCS). For FCS, the studies mainly address the layout of charging stations and the scheduling and management of charging vehicles. Zhu *et al.* [11] proposed a charging scheduling strategy that determines the charging sequence based on the electric vehicles' charging needs rather than their arrival times at the charging stations. Hamed *et al.* [12] considered different charging needs (day and night) and various charging scenarios, formulating the problem as a mixed-integer linear problem with multiple constraints. To address the shortcomings of FCS, such as range anxiety and lengthy charging times, some studies have begun to utilize MCS [13], [14]. Li *et al.* [13] proposed a framework for optimizing mobile charging vehicle operations and developed a variant of a Mixed-Integer Linear Programming (MILP) model. Inspired by the above research, we consider both fixed and mobile scenarios of charging stations, and build the FCS and MCS systems based on the scenario in which UAVs need to travel to charging stations to charge.

The contributions of this paper are summarized as follows.

- Considering the scenario of UAVs traveling to charging stations and the queuing problem caused by limited charging capacity, we model the fixed charging station system and use four different algorithms to solve the problem.
- Based on the FCS system model, the MCS system model is established by adding the mobility of the charging station, and an algorithm to optimize the location of the charging station is proposed. Simulation results indicate that, compared to the FCS, the MCS system model effectively reduces the total flight distance of drones caused by charging.

The rest of the paper is organized as follows. Section II introduces the system model. In Section III, optimization algorithms are introduced. Numerical experiments and analysis are conducted in Section IV. Finally, Section V concludes this paper.

## II. SYSTEM MODEL

In this section, a fixed charging station system is presented to solve the queuing problem encountered in UAV charging. Then, considering the practical needs of mobile charging stations, this paper establishes a mobile charging station system. This system involves the selection of charging station location and the scheduling of UAVs.

### A. Fixed Charging Station System Model

*1) FCS System:* The fixed charging station system consists of a fixed charging station and several UAVs operating tasks in the vicinity of the FCS. Each UAV is designated by the index $i$, $i \in \{1, 2, 3, \ldots, N\}$. The UAVs can communicate with the FCS, sending their current data to the FCS. The FCS is equipped with the capability to collect and process various data from UAVs, enabling it to calculate charging sequence schemes. Data related to UAVs and the charging station are presented in Table I. The FCS has sufficient energy but a fixed number of charging ports. When all charging ports are occupied by UAVs, the remaining UAVs need to queue up in the charging sequence and wait their turn. The number of charging ports is denoted as $M$ and the coordinates of the FCS are located at the origin.

The scheduling process of the FCS system is shown in Fig. 1. A complete charging scheduling process involves: at regular time intervals $T$, the FCS collects relevant data from each UAV, confirms the set of UAVs capable of reaching the FCS, and proceeds with the solution of the scheduling scheme. UAVs unable to reach the FCS do not participate in the schedule and terminate their tasks. At the same time, the participating UAVs move to the FCS for charging. Upon arrival at the station, each UAV lines up at the charging port in the charging sequence provided by the FCS. After finish charging, they will autonomously return to the mission site to continue their missions. When all the UAVs involved in the scheduling return to their mission sites, the scheduling process for this round ends.

TABLE I: Description of the symbol

| Symbol | Description | Unit |
|--------|-------------|------|
| $C_i$ | The battery capacity of the UAV $i$ | Wh |
| $I_i$ | The initial charge of the UAV $i$ | Wh |
| $v_i$ | The flying speed of the UAV $i$ | km/h |
| $x_i$ | The abscissa of the UAV $i$ | km |
| $y_i$ | The ordinate of the UAV $i$ | km |
| $Pf_i$ | The mobile power of the UAV $i$ | W |
| $Pc_i$ | The charging power of the UAV $i$ | W |
| $N$ | The total number of UAVs | / |
| $M$ | The number of charging ports for the charging station | / |

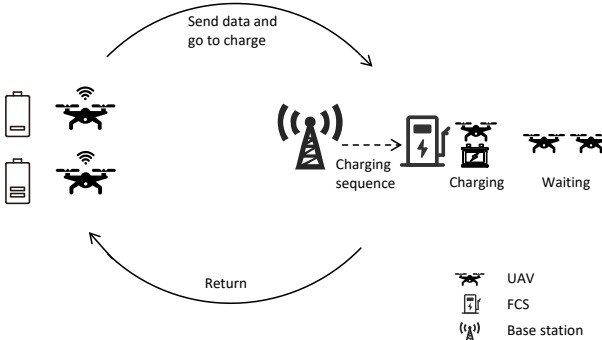

Fig. 1: Scheduling process of FCS system

During the simulation, it is observed that the time required to solve the scheduling solution is significantly shorter than the arrival time of the UAVs. Therefore, it is reasonable to allow the UAVs to travel to the FCS at the beginning of the scheduling process.

*2) Problem Formulation:* We choose the optimization objective as the total duration from the start of the charging schedule until all UAVs return to their respective task points to resume task execution. This aims to minimize the time wasted by UAVs due to charging.

To calculate the charging sequence, the first step is to assess the eligibility of UAVs for scheduling. The time at which UAV $i$ arrives at the charging location is represented as:

$$t_i = \frac{S_i}{v_i}, \tag{1}$$

where $v_i$ represents the flying speed of UAV $i$. $S_i$ is the distance from UAV $i$ to the charging station, where $S_i = \sqrt{x_i^2 + y_i^2}$. The condition that the remaining battery level of UAV $i$ needs to satisfy upon reaching the charging location can be represented as:

$$I_i - t_i Pf_i \geq 0, \tag{2}$$

where $I_i$ is the initial battery level of UAV $i$ and $Pf_i$ is the flight power of UAV $i$. If condition (2) is satisfied, UAV $i$ participates in the current round of scheduling; otherwise, it does not participate. Then, update the value of parameter $N$ to represent the total number of UAVs participating in the scheduling.

At each moment, the number of UAVs currently charging must not exceed the number of charging ports available at the charging point. This constraint can be represented as:

$$\sum_{i=1}^{N} \chi_i \leq M, \tag{3}$$

where $\chi_i$ is a binary function. When UAV $i$ is charging, $\chi_i$ equals 1; otherwise, it equals 0.

The scheduling solution provided by the model is the charging sequence of the UAVs, which is a permutation of the numbers from 1 to $N$. This permutation is denoted as $p$, $p \in P_N$. $P_N$ is the set consisting of all permutations containing the numbers 1 to $N$. $p[j]$ is the $j$-th element in the permutation $p$, representing the UAV index positioned at the $j$-th slot in the permutation. This implies that for $p[j] = k$, $j$ represents the charging order of the UAV with index $k$. For consistency, we denote in the permutation $p$ that $p[j_i] = i$, where $j_i$ represents the charging order of the UAV with index $i$. After UAVs reach the charging location, they charge sequentially according to the charging order. If there are no available charging ports, they have to wait. $Tw_i(p[j_i])$ is the waiting time of UAV $i$ in the permutation $p$. Note that this refers solely to the waiting time of UAV $i$ in the context of permutation $p$, and not as a complex function. Similarly, $Tc_i(p[j_i])$ represents the charging time of UAV $i$ in the permutation $p$. $t_i$ is denoted as $t_i(p[j_i])$ for consistency. The total duration of charging and movement for UAV $i$ in the permutation $p$ can be represented as:

$$t_{sum_i}(p[j_i]) = 2t_i(p[j_i]) + Tw_i(p[j_i]) + Tc_i(p[j_i]), \tag{4}$$

where $t_i(p[j_i])$ needs to be calculated twice because the UAV needs to travel to the charging location and then return to the task location. After determining the charging schedule time for each UAV, taking the maximum value gives the total duration of the scheduling, denoted as:

$$TI(p) = \max_{i \in N} t_{sum_i}(p[j_i]), \tag{5}$$

where $TI(p)$ is the total duration of the scheduling under the permutation $p$. The UAV charging scheduling problem can be formulated as:

$$
\begin{aligned}
\min \quad & TI(p) \\
\text{s.t.} \quad C1: \quad & I_i - t_i(p[j_i])Pf_i \geq 0 \\
C2: \quad & \sum_{i=1}^{N} \chi_i \leq M \\
C3: \quad & p \in P_N,
\end{aligned}
\tag{6}
$$

where $C1$ requires that the UAV $i$ must have sufficient power to reach the charging location. $C2$ requires that the number of UAVs charging simultaneously does not exceed the number of charging ports available at the charging location. $C3$ represents the solution of the problem as a charging order arrangement.

## B. Mobile Charging Station System Model

*1) MCS System:* The mobile charging station system consists of a mobile charging station and several UAVs operating tasks in the vicinity of the MCS. The only difference between the FCS and MCS systems is that the charging location in the former is fixed, while that in the latter is mobile. Therefore, the scheduling problem is divided into two parts: the selection of MCS location and the scheduling of UAVs.

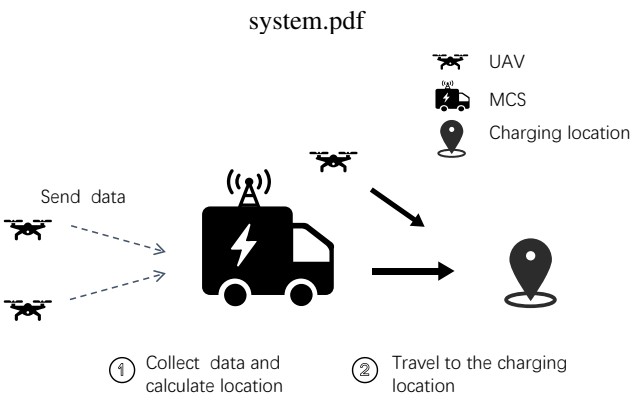

Fig. 2: Scheduling process of MCS system

The scheduling process of the MCS system is shown in Fig. 2. A complete charging scheduling process for the MCS system involves: at regular time intervals $T$, the MCS gathers relevant data from the UAVs to determine the MCS location. Once the location $L$ for the MCS is determined, both the MCS and UAVs capable of reaching location $L$ proceed to $L$ simultaneously. The charging sequence scheme is calculated immediately after selecting location $L$. The round of charging scheduling process is considered complete when all UAVs involved in the charging schedule have returned to their respective mission points. At the beginning of each scheduling round, the coordinates of the MCS are initialized to the origin.

*2) Problem Formulation:* For selecting the MCS location, the ideal location $L$ is characterized by the charging station being able to accommodate more UAVs for scheduling, while the total distance for UAVs to reach the location is shorter. We assume that the selected location is $L = (x_L, y_L)$; the distance from UAV $i$ to location $L$ is $S_i$, where $S_i = \sqrt{(x_i - x_L)^2 + (y_i - y_L)^2}$. For the FCS system, the values of $x_L$ and $y_L$ are always 0. Define the variable $\beta_i$ to indicate whether UAV $i$ participates in scheduling, as follows:

$$
\beta_i = \begin{cases} 1 & \text{if } I_i - t_iPf_i \geq 0 \\ 0 & \text{otherwise,} \end{cases}
\tag{7}
$$

where $I_i - t_iPf_i \geq 0$ indicates that UAV $i$ has enough power to reach the charging location. Then, the location selection problem can be formulated as:

$$\min_{x_L, y_L} \quad -\sum_{i=1}^{N} \beta_i + \alpha \sum_{i=1}^{N} S_i \beta_i, \tag{8}$$

where $\alpha = 0.01$, $\sum_{i=1}^{N} \beta_i$ and $\sum_{i=1}^{N} S_i \beta_i$ represent the total number of UAVs participating in scheduling and the sum of the distances from these UAVs to the charging location. Setting a small value for $\alpha$ ensures that the number of UAVs participating in scheduling is prioritized before optimizing the total distance.

The scheduling after selecting the charging station location is the same as that in the FCS system. Therefore, it will not be elaborated upon further.

---

**Algorithm 1** Selection of MCS location

---

**Initialization:**

1: The abscissa of the UAV $i$ $x_i$, the ordinate of the UAV $i$ $y_i$, the movement speed of the UAV $i$ $v_i$, the battery capacity of the UAV $i$ $C_i$, the initial charge of the UAV $i$ $I_i$, the mobile power of the UAV $i$ $Pf_i$

**Iteration:**

1: Through the battery capacity, the movement speed, the battery capacity, the initial charge and the mobile power of UAVs, calculate the movement radius $r_i$ of UAVs

$$r_i = \frac{C_i - I_i}{Pf_i} v_i$$

2: Take the coordinate $(x_i, y_i)$ of UAVs as the center of the circle, and the movement radius $r_i$ as the radius of the circle to make a circular area

3: Find the area with the most overlapping regions to get the set of candidate coordinates

4: Through traversal or heuristics, select the coordinate point that makes the total distance of the objective function the shortest $\min_{x_L, y_L} - \sum_{i=1}^{N} \beta_i + \alpha \sum_{i=1}^{N} S_i \beta_i$

**Output:** The location $L$

---

### C. Assumptions

1) Measurements of the initial power of the UAV, the coordinates of the UAV relative to the FCS/MCS, and the power consumed by the UAV are accurate.
2) The communication time between the drone and the FCS/MCS is neglected.
3) The time required to change the charging interface for the UAV is neglected.
4) The time for the UAV to reach the charging location is always greater than the time required by the FCS/MCS to determine the scheduling plan.
5) After selecting the charging location, the MCS always arrives at the charging site before the UAVs.

Assumptions 1)–3) are common and weak assumptions on the UAV charging service problem. Simulation results indicate that the computation time for the MCS/FCS scheduling scheme ranges from a few seconds to several minutes, which is significantly shorter than the time required for the drone to reach the charging station. Since the MCS is located roughly in the center of the swarm, the charging location will be relatively close to the initial MCS location. The MCS moves faster than

UAVs, which ensures that it reaches the charging location faster. Therefore, Assumptions 4)–5) are reasonable.

## III. Optimization Algorithms

### A. Scheduling of UAVs

It can be seen that the optimization problem we have established is a variant of the traveling salesman problem. Therefore, heuristic algorithms can be used to solve it. We select simulated annealing algorithm (SA), genetic algorithm (GA), tabu search algorithm (TS), and particle swarm optimization algorithm (PSO). By comparing the performance of four heuristic algorithms, we chose the simulated annealing algorithm. Details can be found in Section IV.

TABLE II: The parameter settings for each algorithm

| Heuristics | Parameter settings |
|---|---|
| PSO | The individual learning factor is $0.5$; the social learning factor is $0.3$; the inertia factor is $1$; the number of particles is $30$; the maximum inertia factor is $1$; the minimum inertia factor is $0.8$. |
| GA | The number of immunized individuals is $30$; the crossover probability is $0.95$; the mutation probability is $0.1$. |
| TS | The number of neighborhood solutions: $N \times (N-1)/2$ for $N \leq 10$, $50$ otherwise; the number of candidate solutions is $25$; the taboo length is $(N \times (N-1)/2)^{0.5}$. |
| SA | The initial temperature is $33 \times N$; the number of cycles in the inner layer is $26 \times N$; the temperature drop rate is $0.98$; accept the new solution with probability $e^{-18 \times \Delta E/T_{\text{init}}}$. |

### B. Selection of MCS location

Based on the objectives of maximizing the number of UAVs involved in scheduling and minimizing the total distance traveled by the UAVs, an algorithm can be designed to solve the problem. First, the circles representing the reachable areas of UAVs are calculated based on UAVs data. Then, the region with the highest overlap of these circles is identified, representing the area accessible to all UAVs. Within this area, the point that minimizes the total distance for UAVs to reach the charging location is determined as location $L$.

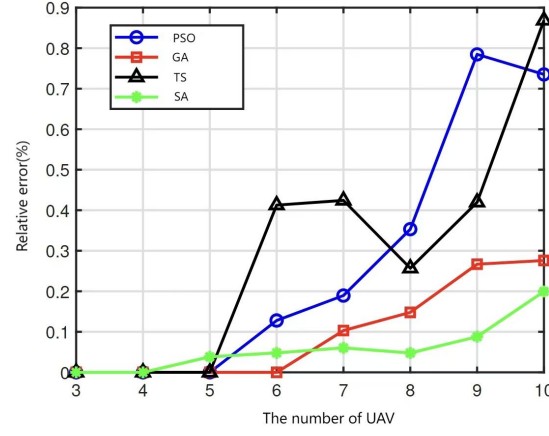

Fig. 3: Comparison of relative errors among four algorithms

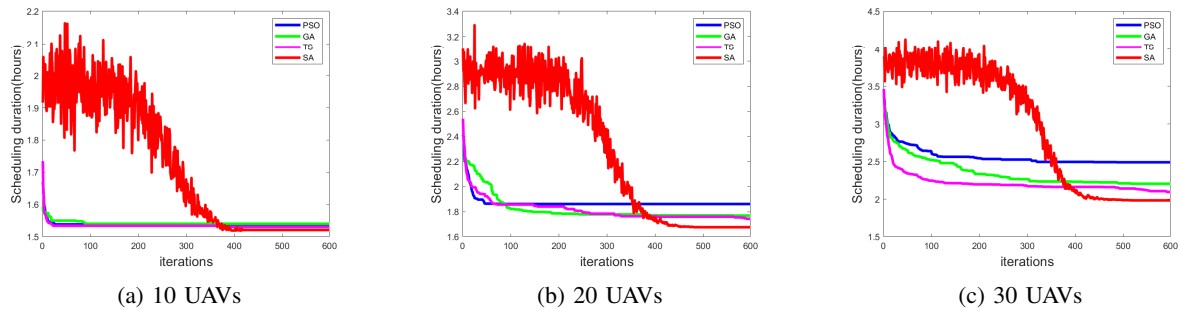

(a) 10 UAVs       (b) 20 UAVs       (c) 30 UAVs

Fig. 4: Comparison of scheduling duration for SA, PSO, TS, and GA with different numbers of UAVs

TABLE III: Comparison of results between FCS and MCS

| System Model | The number of UAVs | Calculation duration | Scheduling duration | The total distance traveled by the UAV | Location coordinates of the charging vehicle |
|---|---|---|---|---|---|
| FCS | 10 | 1.604189s | 1.5147h | 48.1646km | [0, 0] |
| | 15 | 3.705172s | 3.5737h | 62.4658km | [0, 0] |
| | 20 | 5.776012s | 2.2079h | 99.3726km | [0, 0] |
| | 25 | 9.441099s | 3.7836h | 93.0275km | [0, 0] |
| | 30 | 12.593925s | 3.1608h | 144.8695km | [0, 0] |
| MCS | 10 | 1.815212s | 1.4937h | 34.6202km | [−4, 4] |
| | 15 | 3.766389s | 3.4914h | 33.117km | [3, 2] |
| | 20 | 6.124744s | 1.9614h | 77.5759km | [4, 4] |
| | 25 | 9.470387s | 3.7803h | 73.9708km | [0, 2] |
| | 30 | 13.224203s | 3.1273h | 119.9724km | [3, 2] |

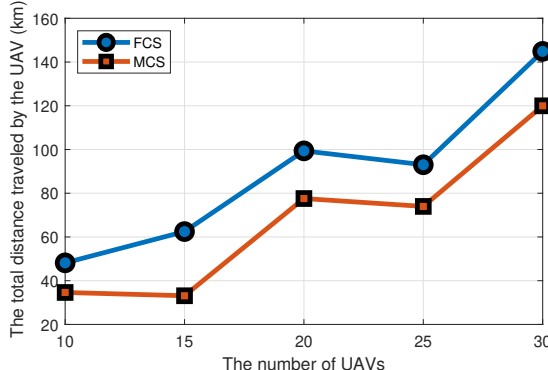

(a) The total distance traveled by UAVs

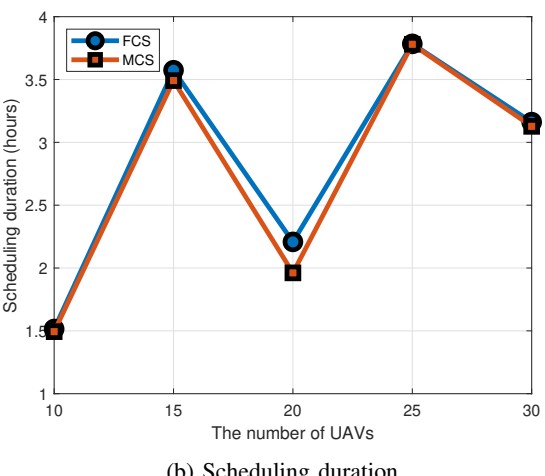

(b) Scheduling duration

Fig. 5: Comparison of UAV distance traveled and scheduling duration for FCS and MCS

To simplify the problem, only integer points on a coordinate grid (points where both the horizontal and vertical coordinates are integers) are considered. Initially, set the value of each integer point to 0. Each time a circular area covers a point, increment the value of that point by 1. By selecting the integer points with the highest values, a finite set of integer points can be obtained as candidate locations. We designate this set as $G$, where the elements in $G$ are the coordinates of points, with elements in the form such as (1, 2). Subsequently, by traversing and solving or employing a heuristic algorithm, the location $L$ can be determined.

The pseudo code for the algorithm addressing this problem is provided in Algorithm 1.

## IV. NUMERICAL EXPERIMENTS AND ANALYSIS

For testing the FCS system model, we set the parameter $M = 2$. Based on simulation experiments, it is found that, even with 30 UAVs, the scheduling time remains within 4 hours. Therefore, we set the scheduling cycle parameter $T = 4$ (which can be adjusted according to the number of drones and the number of charging ports at the charging station). Since the optimal solution can be obtained through exhaustive search when the number of UAVs is 10 or fewer, this part compares the relative error of the solution results of four heuristic algorithms against the optimal solution with the number of UAVs ranging from 3 to 10. The relative error is defined as the difference between the solution obtained by the algorithm and the optimal solution, divided by the optimal solution. The algorithm parameters are set as shown in Table II.

The comparison results of the algorithms are shown in Fig. 3, where the relative errors of all four heuristic algorithms are less than 0.9%. This indicates the effectiveness of the

heuristic algorithms in solving this problem. It is noteworthy that the relative error of SA is generally lower. To compare the performance of heuristic algorithms with a larger number of UAVs, we test the four algorithms with 10, 20, and 30 UAVs, as shown in Fig. 4. Fig. 4 shows that SA consistently outperforms the other three algorithms. This is the reason for selecting SA for this problem.

For testing the MCS system model, we set the parameter $M = 4$ and employ SA for solution. We compare the results of the MCS and FCS systems under the same drone data, as shown in Table III and Fig. 5. It can be seen that, compared to the FCS system, the MCS system results in shorter scheduling times and a significant reduction in the total distance traveled by UAVs. The time required to solve the MCS system is similar to that of the FCS system. This indicates that the MCS system provides a better charging scheduling solution while maintaining similar solution speed, demonstrating the effectiveness and superiority of this model.

## V. Conclusion

A scheduling system with fixed charging station is described. Through simulation, it is concluded that the simulated annealing algorithm is the most suitable method among the four algorithms for solving this problem. Compared to GA, PSO, and TS, SA demonstrates superior performance in solving large-scale drone problems. Based on FCS system, a scheduling system is proposed that replaces fixed charging station with a mobile charging station. For the MCS system, a two-stage optimization model is established, including the selection of MCS location and the scheduling of drone charging. By comparing the results of two models, it is validated that the MCS system model can provide a more optimal scheduling solution to meet practical needs. The future work is to study large scheduling systems with relay charging platforms and to design efficient algorithms for determining charging locations.

## Acknowledgment

This work was supported in part by the National Natural Science Foundation of China under Grant Nos. 62173087. Corresponding author: Zeyu Guo.

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
