# OpenReview forum: "Modeling and analysis of UAV charging scheduling in fixed/mobile charging station systems"
_IEEE.org/ICIST/2024/Conference — IEEE ICIST 2024 Conference Submission_

### Official Review · Reviewer_4vQf · 2024-08-21
**This manuscript has a certain degree of innovation and clear simulation figures. It is recommended to accept this paper for publication in IEEE ICIST 2024.**

**Rating:** 7
**Confidence:** 4

**Review:**

This manuscript has a certain degree of innovation and clear simulation figures. Please answer the following review questions.

You state that a mobile charging vehicle system is established, and the optimization of charging locations is implemented based on the current status of UAVs. What evaluation metrics were used to assess the effectiveness of this optimization? How did the system perform compared to alternative approaches or under different scenarios?

Since both a fixed charging station system and a mobile charging vehicle system are discussed, how do the two approaches compare in terms of their advantages, disadvantages, and suitability for different applications? What factors would influence the choice between the two systems?

---

### Official Review · Reviewer_xcLw · 2024-08-21
**This paper proposes two novel mathematical models for optimizing the charging schedules of Unmanned Aerial Vehicles (UAVs) within systems featuring either fixed or mobile charging stations. Meanwhile, the experiments verified the effectiveness of the method. Here are some comments.**

**Rating:** 7
**Confidence:** 3

**Review:**

Question 1:
Please provide a detailed comparison of the four algorithms evaluated for the fixed charging station (FCS) system. What criteria were used to assess their performance, and what were the strengths and weaknesses of each algorithm?
Question 2:
Please explain the approach used to optimize the mobile charging station’s position. How does this algorithm improve charging efficiency compared to existing methods, and what factors are considered in determining the optimal position?
Question 3:
Please provide more details on the numerical example used to compare algorithm performance. What specific scenarios or parameters were tested, and how do the results demonstrate the effectiveness of the proposed algorithm in improving charging efficiency?

---

### Official Review · Reviewer_yY4y · 2024-08-21
**Manuscript Accept**

**Rating:** 7
**Confidence:** 4

**Review:**

How do the authors choose the best performance in four different algorithms?
Compared with the fixed position of the charging station, does it bring an extra burden when the position of the charging station is changed?
How do the authors measure the effect of optimization?

---

### Decision · Program_Chairs · 2024-09-08

Accept (Oral)